# Synthesis and Characterization of a Novel Two-Dimensional Copper p-Aminophenol Metal–Organic Framework and Investigation of Its Tribological Properties

**DOI:** 10.3390/ma16176061

**Published:** 2023-09-04

**Authors:** Lei Li, Zhijun Liu, Chuan Li, Xiaodong Wang, Mingling Li

**Affiliations:** 1School of Chemistry and Material Engineering, Chaohu University, Hefei 238024, China; 2Engineering Research Center for Preparation and Application of Industrial Ceramic, Chaohu University, Hefei 238024, China; 3Engineering Research Center of High-Frequency Soft Magnetic Materials and Ceramic Powder Materials of Anhui Province, Chaohu University, Hefei 238024, China

**Keywords:** copper, metal–organic framework, two-dimensional, tribological properties

## Abstract

Here, a novel copper p-aminophenol metal–organic framework (Cu(PAP)_2_) is first reported. Powder X-ray diffraction (PXRD), infrared spectra (FTIR), Raman spectra, transmission electron microscopy (TEM) and X-ray photoemission spectroscopy (XPS), in combination with a structure simulation, indicated that Cu(PAP)_2_ is a two-dimensional (2D) material with a staggered structure analogous to that of graphite. Based on its 2D graphite-like layer structure, Cu(PAP)_2_ was expected to exhibit preferable tribological behaviors as an additive in liquid lubricants, and the tribological properties of Cu(PAP)_2_ as a lubricating additive in hydrogenated polydecene (PAO6) or deionized water were investigated. Compared to PAO6 or deionized water, the results indicated that deionized water-based Cu(PAP)_2_ showed much better friction reduction and anti-wear behavior than PAO6-based Cu(PAP)_2_ did, which was due to Cu(PAP)_2_ penetrating the interface between friction pairs in deionized water, but not in PAO6, thus producing lower friction and wear resistance values.

## 1. Introduction

Two-dimensional (2D) metal–organic frameworks (MOFs) refer to layer-stacked MOFs comprising conjugated building blocks and square–planar linkages, with weak Van der Waals interactions between the layers [1]. The 2D MOFs were first reported in 2012, and over ten types of 2D MOFs have been developed based on different ligands and linkages in the last decade [2,3]. Besides the preserved characteristics of traditional MOFs (tunable porosity and versatile structures), 2D MOFs exhibit unique chemical and physical features, such as high stability, electrochemical activity, photoactivity and superior electrical conductivity, and have attracted extensive interest due to their potential applications in energy storage, sensors and photoelectrocatalysis [4,5,6,7,8].

Although 2D MOFs possess a graphite-like, layered structure, they have gained a scarce amount of attention as a lubrication additive. Up until now, the few reports on the tribological properties of MOFs are all based on bulk or nanosheet three-dimensional (3D) MOFs, such as ZIF-8, ZIF-67, ZIF-11, ZIF-71 and MOF-5 [9,10,11,12,13], and the tribological properties of 2D MOFs have not been reported yet. However, the tribological properties of inorganic 2D materials have been extensively studied and exhibit excellent tribological performances, such as graphene, transition metal dichalcogenides (TMDCs), graphitic carbon nitride (g-C_3_N_4_), layered metal oxides, hexagonal boron nitride (h-BN) and black phosphorus (BP) [14,15,16,17,18,19].

Herein, we report the synthesis of a novel 2D copper p-aminophenol metal–organic framework (Cu(PAP)_2_) via a reaction between p-aminophenol hydrochloride and copper chloride in the presence of triethylamine (Figure 1). The structure of Cu(PAP)_2_ was determined via a series of characterizations in combination with density functional tight binding (DFT) calculations, which indicated that Cu(PAP)_2_ has a 2D staggered structure and Cu(II) is partially reduced to Cu(I) by the ligand during synthesis. In Cu(PAP)_2_, copper atoms are coordinated to the amino and phenolic hydroxyl groups of p-aminophenol in a square planar arrangement in the ab plane based on hexagonal unit cells, and interlayer π–π stacking is distributed in the c direction. Furthermore, as a representative of 2D MOFs, the tribological properties of Cu(PAP)_2_ in PAO6 and deionized water were investigated.

## 2. Materials and Methods

### 2.1. Materials

p-Aminophenol hydrochloride, copper chloride and triethylamine were purchased from Macklin Reagent (Shanghai) Co., Ltd., Shanghai, China. Methanol and tetrahydrofuran were purchased from Sinopharm Group, China. Hydrogenated polydecene (PAO6) was purchased from Panhua Chemical (Shanghai) Co., Ltd., Shanghai, China. All the chemical reagents were used as received without further purification. Deionized water was prepared using a water purification system (Sichuan ULUPRE Technology Co., Ltd., Chengdu, China). 

### 2.2. Preparation of Cu(PAP)_2_

The typical preparation procedure for Cu(PAP)_2_ was as follows: p-aminophenol hydrochloride (2.90 g, 0.02 mol) and copper chloride (1.34 g, 0.01 mol) were dissolved in 50 mL of methanol. Triethylamine (4.44 g, 0.044 mol) was added dropwise to the methanol solution while the solution was vigorously stirred. The reaction was kept at room temperature for 1 day in an inert atmosphere. The solid was collected via filtration and washed with methanol and tetrahydrofuran. The obtained solid was dried in a high vacuum at 60 °C for 12 h. 

### 2.3. Preparation of PAO6 or Deionized Water-Based Cu(PAP)_2_

PAO6 or deionized water-based Cu(PAP)_2_ were prepared in the following steps: 0.1 g of Cu(PAP)_2_ was mixed with 20 mL of ethanol, and the mixture was ultrasonicated for 6 h to give a homogeneous dispersion with an obvious Tyndall effect. Then, 1 mL of the Cu(PAP)_2_–ethanol mixture was dispersed into 15 mL of PAO6 or deionized water via an ultrasonic treatment for 1 h to give PAO6-based Cu(PAP)_2_ and deionized water-based Cu(PAP)_2_, respectively. No sedimentation was observed in the PAO6-based Cu(PAP)_2_ and deionized water-based Cu(PAP)_2_ dispersion after 6 h. 

### 2.4. Characterization

Powder X-ray diffraction (PXRD) patterns were recorded using a Rigaku D/MAX2500PCA X-ray diffractometer (Rigaku, Tokyo, Japan) equipped with Cu Kα radiation at 40 kV and 40 Ma. FTIR spectra were characterized using a Gangdong FTIR-650G infrared spectrometer (Gangdong Sci, Tianjin, China). XPS measurements were performed using a Thermo Fisher Scientific 250 XI spectrometer (Thermo Fisher Scientific, Waltham, MA, USA) with Al Ka X-ray as the excitation source. Thermal gravimetric (TG) and differential scanning calorimetry (DSC) were performed using a METTLER TOLEDO TGA/DSC 3+ analyzer under air flow (METTLER TOLEDO, Singapore). Scanning electron microscopy (SEM) was conducted using a Hitachi Regulus 8200 (Hitachi, Tokyo, Japan) with 3 kV accelerating voltage equipped with a Bruker XFlash6 energy dispersive spectroscopy machine (Bruker, Billerica, MA, USA). Raman spectra were analyzed using a Thermo Fisher Scientific DXR2 micro-Raman spectroscopy machine, with laser excitation at 532 nm. Transmission electron microscopy (TEM) was performed using an FEI instrument model Tecnai G20 operating (FEI, Lausanne, Switzerland) at an accelerating voltage of 200 kV. The sample was prepared by drop casting an ethanol dispersion onto TEM grids (Cu mesh), which was then dried in ambient conditions.

### 2.5. Structure Simulation 

The Cu(PAP)_2_ model was generated using the Materials Studio 7.0 suite of programs. Vertex positions were obtained from the Reticular Chemistry Structure Resource (RCSR). The unit cell structure of Cu(PAP)_2_ was calculated using the DFT method. The Coulombic interaction between partial atomic charges was determined using the self-consistent charge (SCC) formalism. The lattice dimensions were optimized simultaneously using a geometry technique. The standard DFTB parameters for X–Y element pair (X, Y = C, H, O, N, Cu) interactions were selected from the matsci set. Pawley refinement was carried out using Reflex, a software package 7.0 for crystal determination from XRD patterns. Peak-broadening (pseudo-Voigt function), asymmetry correction (Berrar–Baldinozzi function) and zero-shift errors were refined together to achieve the improved profile fitting [20]. 

### 2.6. Tribological Tests

The test and analysis methods for the tribological properties of PAO6 or deionized water-based Cu(PAP)_2_ are shown in Figure 2. The tribological test was carried out using the China-made HSR-2 M reciprocating friction and wear tester (Lanzhou Zhongkekaihua Technology Development Co., Ltd., Lanzhou, China) and friction pairs using ball and disk contact. The test conditions were as follows: rated load: 10 N; sliding speed: 50 mm/s; test time: 30 min. The steel ball and steel disk used in the test were made of GCr15 steel, the diameter of the ball was ϕ6 mm, the hardness was HV 647-861 and the surface roughness was Ra 0.016 μm. The disk was polished before the experiment. The diameter of the disks was ϕ28 × 2 mm, the hardness was HV 300, and the surface roughness was Ra 0.02 μm. The steel ball and steel disk were ultrasonically cleaned with acetone for 15 min before and after the test, and 0.5 mL of the sample was dropped on the contact area of the specimens for each tribological test. The coefficient of friction was automatically recorded using a computer, and the wear volume of the discs was measured via three-dimensional laser-scanning microscopy (VK-X100, Keyence, Tokyo, Japan). The material compositions on the wear surface were detected and analyzed as per the Raman spectra.

## 3. Results and Discussion

### 3.1. Structural Analysis of Cu(PAP)_2_

#### 3.1.1. PXRD Analysis and Structure Simulation

The PXRD analysis of Cu(PAP)_2_ revealed a crystalline structure with prominent peaks at 2θ = 5.86°, 11.69° and 17.23°, which are indicative of a long-range order within the ab plane. An additional peak at 2θ = 24.53°, corresponding to the (002) reflection, indicated structural ordering that is perpendicular to the layers, with an interlayer distance of 3.63 Å (Figure 3a). To determine the structural model of Cu(PAP)_2_, two possible extended structures based on hexagonal unit cells, eclipsed (AA stacking) and staggered (AB stacking) (Figure 3b), were generated. The geometry of the two models was optimized via a DFT calculation, and the simulated PXRD patterns were calculated (Figure 3a). The AB stacking mode after Pawley refinement was in good agreement with the experimental data, with a low Rwp of 1.84% and an Rp of 1.42% (Figure 4). The unit cell belonged to a P21/M space group with the following lattice parameters: a = 18.197 Å, b = 7.215 Å, c = 18.846 Å and α = γ = 90°, β = 123.74°. All of these results show that Cu(PAP)_2_ is a 2D structure with the AB stacking model.

#### 3.1.2. FTIR and Raman Spectra Analysis

The FTIR spectra of Cu(PAP)_2_ showed a peak at approximately 1486.8 cm^−1^ that can be assigned to a C=C stretching band, while the peak at approximately 1263.2 cm^−1^ can be assigned to overlapped C-N and C-O stretching bands [21] (Figure 5a). All of these bands were significantly different from those of p-Aminophenol hydrochloride, suggesting the formation of a metal–organic framework via the coordination of copper atoms with p-Aminophenol hydrochloride.

The Raman spectra of Cu(PAP)_2_ revealed two peaks at approximately 1334.3 and 1501.8 cm^−1^ (Figure 5b), analogous to the D and G bands of 2D graphitic materials, respectively [22]. While the D band was attributed to edges, defects and disordered carbons, the G band resulted from the vibrations of ordered the sp^2^ carbon in the two-dimensional hexagonal lattice [23]. The results also confirmed that Cu(PAP)_2_ is a 2D structure material.

#### 3.1.3. Morphology and XPS Analysis 

The morphology of Cu(PAP)_2_ is composed of bulk particles with sizes of several micrometers (Figure 6a). Furthermore, the microscopic properties of Cu(PAP)_2_ dispersed by ethanol were characterized via TEM (Figure 6b), which showed the layered stacking and edge folding or curling that is typical of two-dimensional materials. The EDS mapping results indicated that the elements C, O, N and Cu were uniformly distributed in Cu(PAP)_2_ (Figure 6c).

XPS was employed to probe the chemical states of surface elements in Cu(PAP)_2_. The survey spectra revealed the existence of C, N, O and Cu (Figure 7a). In the high-resolution XPS C 1s spectra, the binding energy (BE) peaks located at 283.8, 285.3 and 287.2 eV are attributed to -C=C-, -C-O-(-C-N-) and -C=O(-C=N-) (Figure 7b), respectively. In the O 1s spectra, the deconvoluted peaks at 530.6, 531.8 and 532.7 eV are attributed to -O-Cu(Ⅰ, Ⅱ), -O-C- and O=C- (Figure 7c), respectively. The N 1s spectra also could be deconvoluted into three peaks at 397.6, 399.1 and 400.5 eV, which were assigned to -N-Cu(Ⅰ, Ⅱ), -N-C- and NH=C- (Figure 7d), respectively. In the high-resolution Cu 2p XPS spectra of Cu(PAP)_2_, two signals with binding energies of 932.1 and 934.1 eV were assigned to Cu(I) 2p_1/2_ and Cu(II) 2p_3/2_ [24] (Figure 7e), respectively, while the satellite peaks located at 942.5 and 961.8 eV were likely due to the shakeup excitation of the high-spin Cu(II) states. The results of the XPS analysis correlated well with the proposed Cu(II) being partially reduced to Cu(I) by the ligand during synthesis.

#### 3.1.4. Thermal Stability Analysis

The thermal stability of Cu(PAP)_2_ was analyzed using Thermalgravimetric (TG) and differential scanning calorimetry (DSC). One obvious weight loss process occurred in the thermogram profile (Figure 8). The weight loss that occurred between 210 °C and 380 °C was approximately 76.56%, and this resulted from the decomposition of Cu(PAP)_2_ and the further oxidation of the residue under the air flow, accompanied by two strong endothermic peaks at 218 °C and 248 °C in the DSC curve. The results showed that Cu(PAP)_2_ had good thermal stability in air.

### 3.2. Tribological Properties of Cu(PAP)_2_

#### 3.2.1. Results of Friction and Wear Performance Tests

In order to study the tribological properties of Cu(PAP)_2_, two different dispersions (PAO6 and deionized water) were employed in the test. The friction coefficients of PAO6-based Cu(PAP)_2_ and deionized water-based Cu(PAP)_2_ were both decreased relative to those of PAO6 and deionized water (Figure 9a). As can be observed, there was an obvious difference in the coefficient of friction curves when adding Cu(PAP)_2_ to PAO6 or deionized water. In the test, the coefficient of friction curves of PAO6-based Cu(PAP)_2_ tended to converge to PAO6. However, the coefficient of friction curves of deionized water-based Cu(PAP)_2_ were significantly lower than those of the deionized water. The average friction coefficients of PAO6, PAO6-based Cu(PAP)_2_, deionized water and deionized water-based Cu(PAP)_2_ were also calculated by the tester (Figure 9b), and the average friction coefficients decreased by 14.9% and 35.6% for PAO6-based Cu(PAP)_2_ and deionized water-based Cu(PAP)_2_, which were compared to those of PAO6 and deionized water, respectively.

The 3D worn surface topography and 2D cross-section profiles of the lower plates lubricated with PAO6, PAO6-based Cu(PAP)_2_, deionized water and deionized water-based Cu(PAP)_2_ were obtained using a VK Analyzer (Figure 10a). Compared to PAO6 and deionized water, it was clear that the average wear volume of PAO6-based Cu(PAP)_2_ and deionized water-based Cu(PAP)_2_ showed contrary results (Figure 10b). The average wear volume of the PAO6-based Cu(PAP)_2_ increased by 22.0% relative to that of PAO6; nevertheless, that for the deionized water-based Cu(PAP)_2_ decreased by 23.9% relative to that of the deionized water. The above results confirm that Cu(PAP)_2_ exhibits much better anti-wear behavior in deionized water than it does in PAO6.

#### 3.2.2. Worn Surface Analysis of Raman Spectra

It is generally believed that the friction reduction mechanism of 2D materials based on less shearing is caused by weaker Van der Waals forces between two adjacent layers [25,26,27]. After 2D materials penetrate the interface between friction pairs, the shearing stress of friction can act on the 2D materials, and the interlayer of the 2D layered materials yields a sliding system with a lower force on the contact region of the friction pairs, thus producing less friction and wear resistance [28]. For the sake of explaining the different tribological properties of Cu(PAP)_2_ dispersed in PAO6 and deionized water, Raman spectra of the wear traces were detected (Figure 11). The characteristic peaks of α-Fe_2_O_3_ (292.6 cm^−1^, 411.2 cm^−1^, 222.2 cm^−1^ and 1319.5 cm^−1^) and Fe_3_O_4_ (660.8 cm^−1^) were all observed in the Raman spectra of wear traces lubricated with PAO6 and PAO6-based Cu(PAP)_2_ in relation to steel plate oxidation in the test [29]. However, the two typical peaks (D and G bands) of Cu(PAP)_2_ were not observed in the wear traces with PAO6-based Cu(PAP)_2_ used as a lubricant. Otherwise, the characteristic peaks of Fe_3_O_4_ were observed in the Raman spectra of the wear traces lubricated with deionized water and deionized water-based Cu(PAP)_2_, and additional D (1335.8 cm^−1^) and G (1503.6 cm^−1^) bands of Cu(PAP)_2_ could clearly be observed in the wear traces with deionized water-based Cu(PAP)_2_ used as a lubricant. The results indicated that for the PAO6-based Cu(PAP)_2_, Cu(PAP)_2_ cannot infiltrate the friction pairs in PAO6, which results in a poor tribological performance. For the deionized water-based Cu(PAP)_2_, Cu(PAP)_2_ can smoothly enter the friction pairs to bear the applied load and prevent damage to the steel plates, which leads to much better friction reduction and anti-wear behaviors.

In comparison to inorganic 2D layered materials (graphene, GO, h-BN and MoS_2_), the friction improvement ability of Cu(PAP)_2_ was partially satisfactory, and it displayed preferable anti-friction and certain wear resistance properties (Table 1). The interlayer distance of Cu(PAP)_2_ was 3.63 Å, which was bigger than those of graphene (3.33 Å), MoS_2_ (3.49 Å), h-BN (3.33 Å) and MoS2 (3.33 Å), while it was smaller than that of GO (6.42 Å–9.86 Å). A smaller or bigger interlayer distance implies stronger or weaker Van der Waals forces, respectively, which result in preferable anti-friction behaviors for Cu(PAP)_2_ compared to those of graphene, MoS_2_ and h-BN. However, due to the porous framework structure of MOFs, the mechanical properties of Cu(PAP)_2_ are inferior to those of most inorganic 2D layered materials, which are likely to be responsible for the limited anti-wear improvement.

## 4. Conclusions

In this study, Cu(PAP)_2_ was synthesized via a reaction between p-Aminophenol hydrochloride and copper chloride. Its structure was characterized using powder X-ray diffraction, FTIR spectra, Raman spectra and X-ray photoemission spectroscopy, in combination with structure simulations. Furthermore, the tribological performance of Cu(PAP)_2_ as a lubricating additive in PAO6 or deionized water was investigated using a ball-on-plate tribometer. The conclusions are as follows:First, a novel copper–organic framework was reported. The characterizations and DFT calculation indicated that Cu(PAP)_2_ is a typical 2D material with a staggered structure analogous to that of graphite.Significantly different tribological performances were observed when using Cu(PAP)_2_ as a lubricating additive in PAO6 or deionized water, and deionized water-based Cu(PAP)_2_ showed much better friction reduction (35.6% decrease) and anti-wear (23.9% decrease) behaviors than those of PAO6-based Cu(PAP)_2_.The Raman spectral analysis of the worn surface showed that Cu(PAP)_2_ can penetrate the interface between friction pairs in deionized water, but not in PAO6, which results in the significantly different tribological properties of PAO6-based Cu(PAP)_2_ and deionized water-based Cu(PAP)_2_.

## Figures and Tables

**Figure 1 materials-16-06061-f001:**
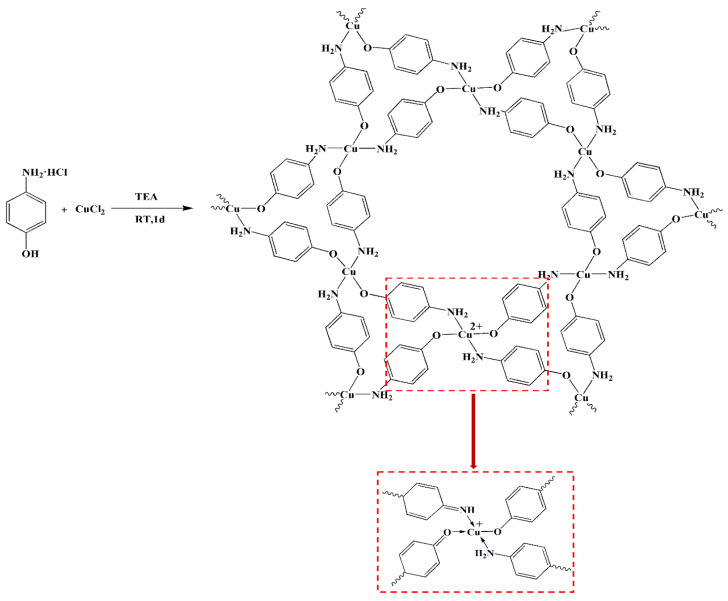
Synthesis of Cu(PAP)_2_. Note that Cu(II) was partially reduced to Cu(I) by the ligand during synthesis.

**Figure 2 materials-16-06061-f002:**
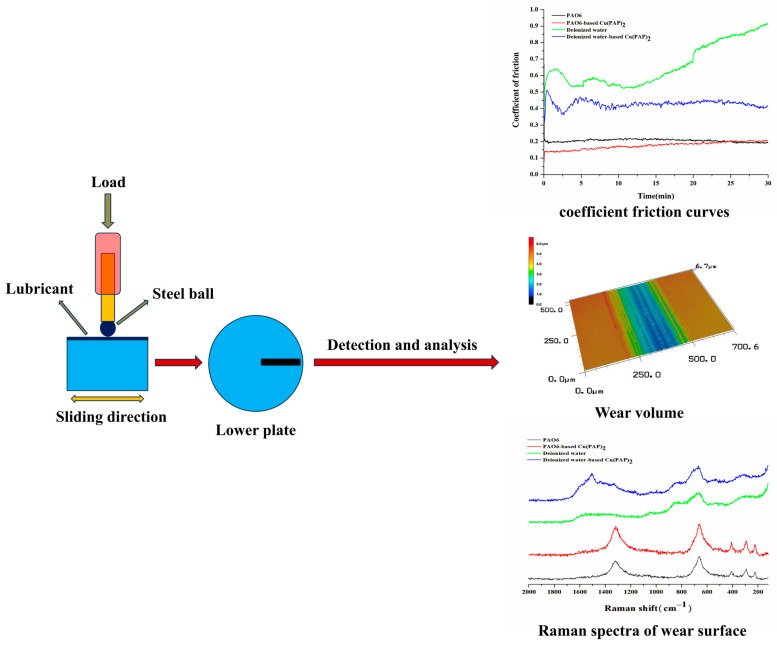
Schematic of the tribological tests and analysis methods.

**Figure 3 materials-16-06061-f003:**
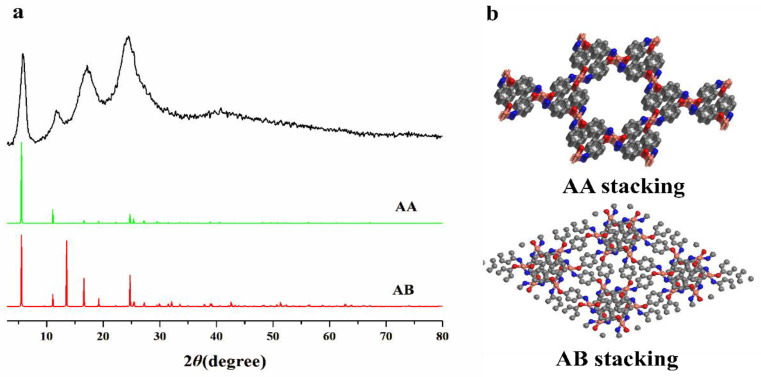
(**a**) Powder XRD patterns of Cu(PAP)_2_: obtained powder (black curve) and simulations using AA (green curve) and AB (red curve) stacking. (**b**) Structural representations of the unit cells of Cu(PAP)_2_ in AA and AB stacking, as viewed from the c axis (gray: carbon; red: oxygen; blue: nitrogen; orange: copper; hydrogen atoms are omitted for clarity).

**Figure 4 materials-16-06061-f004:**
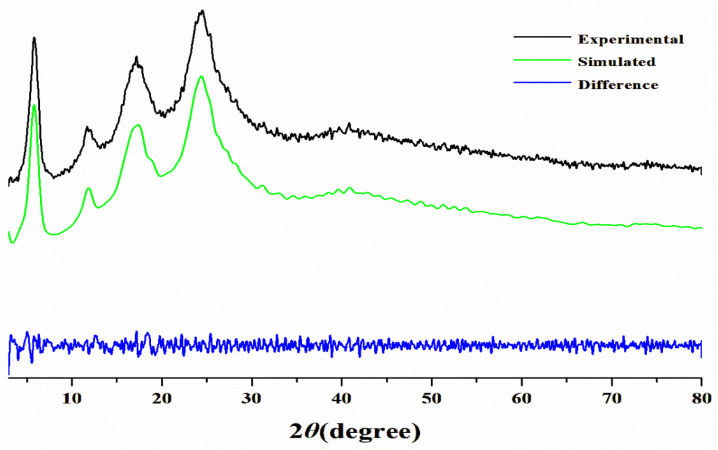
PXRD pattern of Cu(PAP)_2_ with the experimental pattern shown in black, the calculated pattern for the AB stacking model shown in green, and the difference plot shown in blue.

**Figure 5 materials-16-06061-f005:**
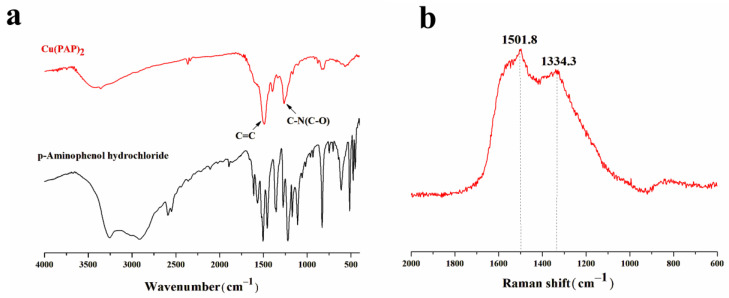
(**a**) FTIR spectra of Cu(PAP)_2_ and p-Aminophenol hydrochloride. (**b**) Raman spectra of Cu(PAP)_2._

**Figure 6 materials-16-06061-f006:**
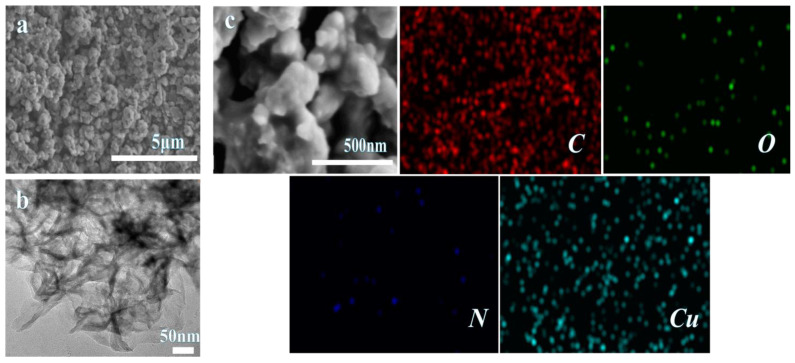
(**a**) SEM images of Cu(PAP)_2_ bulk. (**b**) TEM images of Cu(PAP)_2_ dispersed by ethanol. (**c**) EDS mapping of Cu(PAP)_2_.

**Figure 7 materials-16-06061-f007:**
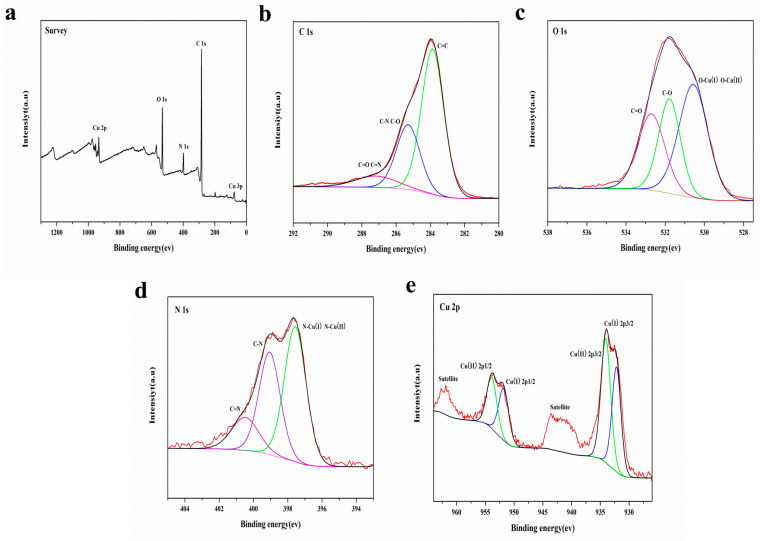
(**a**) XPS survey of Cu(PAP)_2_. (**b**–**e**) High-resolution C 1s, O 1s, N 1s and Cu 2p XPS of Cu(PAP)_2_.

**Figure 8 materials-16-06061-f008:**
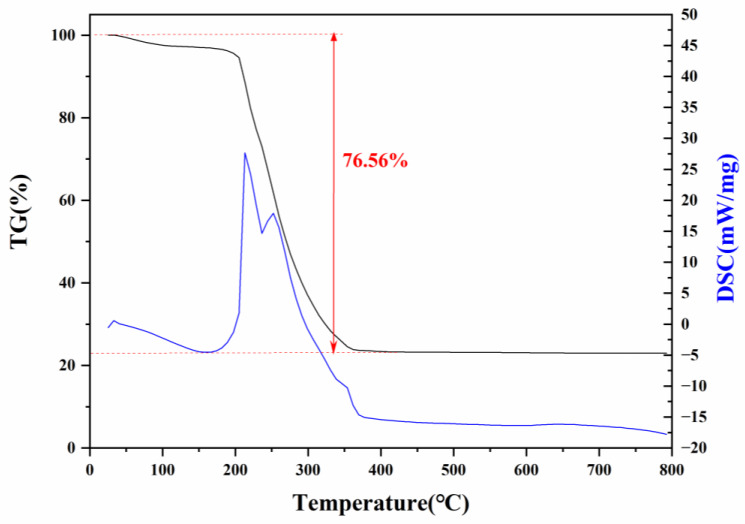
TG and DSC curves of Cu(PAP)_2_ under flowing air.

**Figure 9 materials-16-06061-f009:**
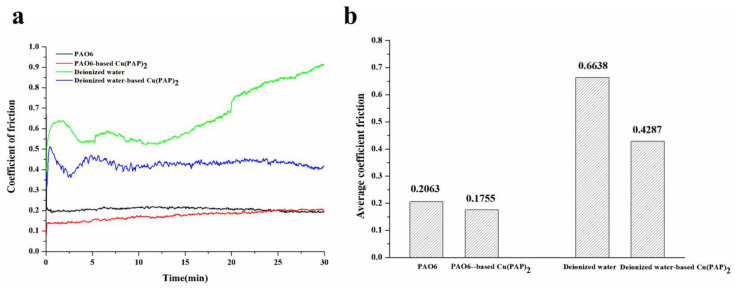
(**a**) The coefficient of friction curves and (**b**) average coefficient of friction for PAO6, PAO6-based Cu(PAP)_2_, deionized water and deionized water-based Cu(PAP)_2_.

**Figure 10 materials-16-06061-f010:**
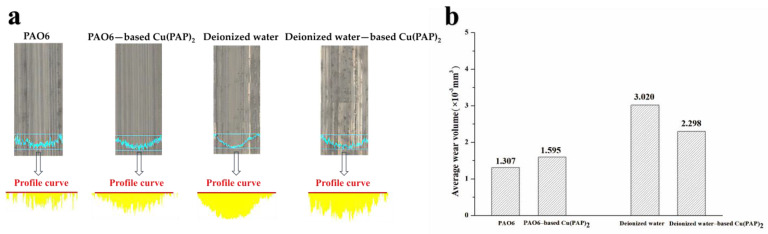
(**a**) Three-dimensional worn surface topography and two-dimensional cross-section profiles of lower plates after tribological testing and (**b**) average wear volumes for PAO6, PAO6-based Cu(PAP)_2_, deionized water and deionized water-based Cu(PAP)_2_.

**Figure 11 materials-16-06061-f011:**
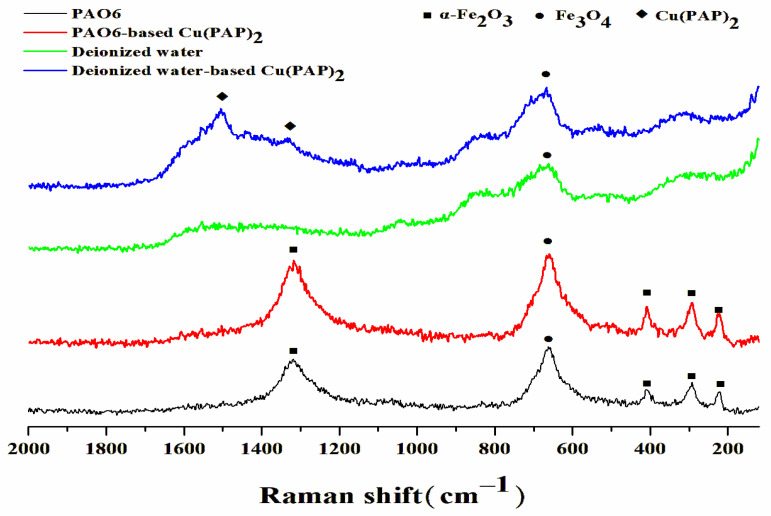
Raman spectra of wear traces of lower plates after tribological testing for PAO6, PAO6-based Cu(PAP)_2_, deionized water and deionized water-based Cu(PAP)_2._

**Table 1 materials-16-06061-t001:** Comparisons of friction performances of 2D materials in water.

2D Materials	Testing Conditions	Tribological Performance		Reference
		Friction Coefficients	Wear	
Cu(PAP)_2_	Ball-on-plate reciprocating tribometer, 10 N	35.6%	23.9%	This work
Graphene	Ball-on-plate reciprocating tribometer, 3 N	21.9%	13.5%	[30]
Graphene Oxide (GO)	Ball-on-plate reciprocating tribometer, 10 N	55.6%	41.6%	[31]
Hexagonal boron nitride (h-BN)	Ball-on-plate reciprocating tribometer, 3 N	22.5%	16.9%	[32]
Molybdenum disulfide (MoS_2_)	Ball-on-plate reciprocating tribometer, 8 N	26.5%	74.6%	[33]

## Data Availability

Not applicable.

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
