# Peer review of "Synthesis and Characterization of a Novel Two-Dimensional Copper p-Aminophenol Metal–Organic Framework and Investigation of Its Tribological Properties"

_materials, 2023, doi:10.3390/ma16176061_

Round 1

Reviewer 1 Report

In this manuscript, the authors reported a Cu-based MOF 2D material. The structure of this MOF material was simulated by DFT calculation and experimental analysis such as XRD, FTIR, and Raman spectra. The authors also investigated the tribological property of the synthesized Cu(PAP)2 material as lubricant in different solvents. The topic of this manuscript is interesting. However, my main concern is whether the obtained Cu(PAP)2 particle can be called a 2D material given that only SEM morphologies were shown. The authors are suggested to address this question. 

Here are some comments.   

1.     The Introduction section of this manuscript needs improvement. The authors are suggested to discuss more about the synthesis, structure, and properties of MOF materials.

2.     To improve the clarity of Figure 2, the authors are suggested to provide the legend of each curve.

3.     The authors showed the Raman spectroscopy of Cu(PAP)2. What is the ratio of the intensity of D and G bond in Figure 5b?

4.     The authors used DFT calculation to confirm that Cu(PAP)2 was a typical 2D material with staggered structure analogous to graphite. They also showed the SEM morphologies of Cu(PAP)2 particles. However, the result is not convincing enough. To confirm whether the obtained Cu(PAP)2 is 2D shape or not, the authors are suggested to provide the TEM image of Cu(PAP)2. Besides, the scale bar in Figure 6 is not clear.

5.     Most Cu-based MOFs are not stable when exposed to air and moisture. Are there open metal sites in this Cu(PAP)2 material? When used in deionized water, will the structure of Cu(PAP)2 still be stable?  

Author Response

Dear Reviewer:

Thank you for your comments concerning our manuscript entitled “Synthesis and Characterization of a Novel Two-Dimensional Copper p-Aminophenol Metal-Organic Framework and Investigation of Its Tribological Properties” (Manuscript ID: materials-2576155). Those comments are valuable and very helpful for revising and improving our paper, as well as the important guiding significance to our researches. We have studied comments carefully and have made correction which we hope meet with your approval. Revised or new portions have been included in the resubmitted manuscript. The following is a point-to-point response to your comments.

1.The introduction section of this manuscript needs improvement. The authors are suggested to discuss more about the synthesis, structure, and properties of MOF materials.

Reply: Thank you for your kind suggestion. As suggested, we have modified the introduction of manuscript and explicated the structure and properties of 2D MOFs. Meanwhile, the mainly synthesis methods of 2D MOFs are simple stirring or solvothermal, which can be easily obtained in relevant references of this manuscript.

2.To improve the clarity of Figure 2, the authors are suggested to provide the legend of each curve.

Reply: Thanks very much for careful guidance. We have provided the legend of each curve in Figure 2 to illustrate the analysis methods for tribological properties of PAO6 or deionized water-based Cu(PAP)2.

3.The authors showed the Raman spectroscopy of Cu(PAP)2. What is the ratio of the intensity of D and G band in Figure 5b?

Reply: We greatly appreciate your insightful comment. The ID/IG value in Figure 5b is 0.84, which indicate that although Cu(PAP)2 is a two-dimensional material, it has many structural defects.

4.The authors used DFT calculation to confirm that Cu(PAP)2 was a typical 2D material with staggered structure analogous to graphite. They also showed the SEM morphologies of Cu(PAP)2 particles. However, the result is not convincing enough. To confirm whether the obtained Cu(PAP)2 is 2D shape or not, the authors are suggested to provide the TEM image of Cu(PAP)2. Besides, the scale bar in Figure 6 is not clear.

Reply: Thank you for your kind suggestion. We agreed with the comment and the microscopic characterization of Cu(PAP)2 dispersed by ethanol was conducted by TEM, which showed the layered stacking and edge folding or curling of typical two-dimensional materials (see Figure 6b in resubmitted manuscript). Meanwhile, the Figure 6 has been revised to improve the quality and clarity.

5.Most Cu-based MOFs are not stable when exposed to air and moisture. Are there open metal sites in this Cu(PAP)2 material? When used in deionized water, will the structure of Cu(PAP)2 still be stable?

Reply: Thank you for your significant reminding. The traditional 3D copper-based MOFs are usually not stable in moisture as the structure may be degraded by hydrolysis. However, although open metal sites are defined, the 2D MOFs has high stability due to its square-planar coordination with high in-plane conjugation. Actually, most 2D copper-based MOFs was synthesize in aqueous solution, such as Cu3(HHTP)2, Cu3(HITP)2, THQ-Cu-MOF, Cu3(TABTO)2, which showed they are stable in water. The coordination model of Cu(PAP)2 is analogous to Cu3(TABTO)2 (see Reference 21 in resubmitted manuscript) and the characteristic peaks in Raman spectra of Cu(PAP)2 were clearly observed in wear traces with deionized water-based Cu(PAP)2 after tribological test, it indicated that Cu(PAP)2 was stable in deionized water.

We have tried our best to improve the manuscript and made some changes in the manuscript.  These changes will not influence the content and framework of the paper. We appreciate your warm work earnestly, and hope that the correction will meet with approval. Once again, special thanks to you for your comments and suggestions.

Reviewer 2 Report

The paper by Li et al. report the synthesis of a new copper p-aminophenol metal organic framework. This novel material has been characterized with a number of techniques such as powder X-ray diffraction, FTIR, Raman and XPS. Moreover the authors also report a DFT-TB optimized geometry. The tribological properties (friction coefficient, wear volume) when used as additive in H2O (deionized) and PAO6 have also been reported. It is found that the tribological performance of this new compound is inferior to that provided by well known inorganic 2D layered materials. 

I think that the paper provides sufficiently new material for publication in Materials. The material's characterization is extensive and, however disappointing the results, they warrant proper documentation. In my opinion the paper should be published after extensive language editing 

In many instances the quality of the language is rather poor. The authors should carefully check the language and improve the general presentation. The quality of several Figures (6, 9) is rather poor.

Author Response

Dear Reviewer:

Thank you for your comments concerning our manuscript entitled “Synthesis and Characterization of a Novel Two-Dimensional Copper p-Aminophenol Metal-Organic Framework and Investigation of Its Tribological Properties” (Manuscript ID: materials-2576155). Those comments are valuable and very helpful for revising and improving our paper, as well as the important guiding significance to our researches. We have studied comments carefully and have made correction which we hope meet with your approval. Revised or new portions have been included in the resubmitted manuscript. The following is response to your comments.

Comments:

In many instances the quality of the language is rather poor. The authors should carefully check the language and improve the general presentation. The quality of several Figures (6, 9) is rather poor.

Reply: Thank you for your suggestions. We apologize for the poor language of our manuscript. According to your advice, this manuscript was edited for proper English language, grammar, punctuation, spelling, and overall style by one of the highly qualified native English speaking editor at MDPI. Meanwhile, Figure 6 and Figure 9 have been revised to improve the quality and clarity.

We have tried our best to improve the manuscript and made some changes in the manuscript.  These changes will not influence the content and framework of the paper. We appreciate your warm work earnestly, and hope that the correction will meet with approval. Once again, special thanks to you for your comments and suggestions.

Reviewer 3 Report

Dear Editor and author

The paper presents synthesis and characterisation of Cu(PAP)2. Both synthetisation and characterisation are well described and well presented. The data are of good quality and the conclusions are sound. The data from tribological tests are well done and presented. All in all the paper is good and should be published. Just some small comments; the figure captions should contain more details in order to understand the figures. For example figure 2 contains a set of graphs with different colors, but it is not well described whta the colors represent. Also, do figure 2 and 4 share the same graphs? It seems a bit unnecessary to present the same data twice. 

Author Response

Dear reviewer:

Thank you for your comments concerning our manuscript entitled “Synthesis and Characterization of a Novel Two-Dimensional Copper p-Aminophenol Metal-Organic Framework and Investigation of Its Tribological Properties” (Manuscript ID: materials-2576155). Those comments are valuable and very helpful for revising and improving our paper, as well as the important guiding significance to our researches. We have studied comments carefully and have made correction which we hope meet with your approval. Revised or new portions have been included in the resubmitted manuscript. The following is response to your comments.

Comments:

The paper presents synthesis and characterization of Cu(PAP)2. Both synthetisation and characterization are well described and well presented. The data are of good quality and the conclusions are sound. The data from tribological tests are well done and presented. All in the paper is good and should be published. Just some small comments; the figure captions should contain more details in order to understand the figures. For example figure 2 contains a set of graphs with different colors, but it is not well described what the colors represent. Also, do figure 2 and 4 share the same graphs? It seems a bit unnecessary to present the same data twice. 

Reply: Thank you for your suggestions. We have provided the legend of each curve in Figure 2 to avoid confusion in understanding. Figure 2 and Figure 4 do not share the same graphs, Figure 2 is schematic of the test and analysis methods for the tribological properties of PAO6 or deionized water-based Cu(PAP)2. However, Figure 4 is the comparison between simulation data of Cu(PAP)2 in AB stacking mode and experimental PXRD pattern.

We have tried our best to improve the manuscript and made some changes in the manuscript.  These changes will not influence the content and framework of the paper. We appreciate your warm work earnestly, and hope that the correction will meet with approval. Once again, special thanks to you for your comments and suggestions.

Round 2

Reviewer 1 Report

The revised manuscript looks good to me. I agree that the manuscript can be accepted.